

# A short scale for measuring attitudes towards the doctor-patient relationship: psychometric properties and measurement invariance of the German Patient-Practitioner-Orientation Scale (PPOS-D6)

Roman Pauli[1,2] and Saskia Wilhelmy[1]

[1] Institute of History, Theory and Ethics of Medicine, RWTH Aachen, Aachen, Germany
[2] Current affiliation: Institute for Occupational, Social and Environmental Medicine, RWTH Aachen, Aachen, Germany

## ABSTRACT

**Background**. The Patient-Practitioner Orientation Scale (PPOS) was originally developed to compare doctor's and patient's consensus regarding patient centeredness. Research assumed PPOS measurements to be comparable across different groups of participants, however, without assessing the actual validity of this assumption. In this study, we investigate the psychometric properties and measurement invariance of a short version of the German translation of the PPOS.

**Methods**. Based on a cross-sectional survey of $N = 332$ medical students, we present a short version of the German Patient-Practitioner-Orientation Scale (PPOS-D6) and examine its psychometric properties as well as measurement invariance across participants with varying levels of medical experience and gender using multigroup confirmatory factor analyses.

**Results**. Results indicate that PPOS-D6 provides valid and reliable measurements of patient-centeredness that are invariant across participants with different medical experience. Preliminary results also suggest invariance across gender.

**Conclusion**. PPOS-D6 is a suitable and efficient measure to compare group-specific attitudes towards the doctor-patient interaction. Additional research on convergent and discriminant validity and divergent study samples is advised.

Corresponding author
Roman Pauli, rpauli@ukaachen.de

## INTRODUCTION

The doctor-patient relationship is an intimate situation in which a person reveals vulnerability to another in hope of healing or help (*Gordon, Phillips & Beresin, 2010*). In this context, four (ideal) models of interaction between doctor and patient can be distinguished: the paternalistic, the deliberative, the interpretive and the informative model (*Emanuel & Emanuel, 1992*). The concept of shared decision-making provides a mediating

role between the two poles of "paternalistic" and "informative" (*Elwyn, Edwards & Kinnersley, 1999*), which aims, *e.g.*, to ensure patient autonomy and to make joint decisions (*Bomhof-Roordink et al., 2019*). Patient-centeredness has become a crucial supplement to the bio-medical view, associated, *e.g.*, with improved physical health outcomes or efficiency of care (*Rathert, Wyrwich & Boren, 2013*; *Robinson et al., 2008*; *Michie, Miles & Weinman, 2003*; *Mead, Bower & Hann, 2002*; *Stewart et al., 2000*).

In order to find out to what extent doctors and patients coincide in their assessments of a treatment interaction, *Krupat et al. (2000)* developed the Patient Practitioner Orientation Scale (PPOS). The scale measures whether patients and practitioners are rather patient- or practitioner-centered in their attitudes and in how far they agree in their preferences. The PPOS has been translated into numerous languages (*e.g.*, *Hurley et al., 2018*; *Wang et al., 2017*; *Pereira et al., 2013*) and used to compare patient-centeredness across different audiences or associations with health outcomes (*Ahmad et al., 2018*). Previous research implicitly assumed that PPOS measurements are comparable across different groups. However, this theoretical assumption was never tested for empirical evidence. Therefore, we want to draw attention to the concept of measurement invariance as a prerequisite for group comparisons of latent constructs. In addition, we want to comply with the high demand for short scales in settings such as clinical practice (*Ziegler, Kemper & Kruyen, 2014*). Prior research showed that questionnaire length is negatively associated with response rates (*Edwards et al., 2004*). Considering that time is a crucial and limited resource in clinical practice, short scales allow a practicable and valid measurement of the constructs of interest (*Rammstedt & Beierlein, 2014*). The aims of our study are to introduce a short version of the German translation of the PPOS and to investigate its psychometric properties as well as measurement invariance across participants with and without medical experience and gender.

## Measuring the doctor-patient-relationship

Several psychometric scales concerned with operationalizing the doctor-patient relationship represent different dimensions of interaction (*e.g.*, empathy or therapeutic alliance; see *Eveleigh et al., 2012*). The PPOS was developed to model attitudes towards patient-centeredness across doctors and patients. This was operationalized by 18 items as indicators for the extent of patient orientation in therapy decisions. The scale assumed that patient orientation is a two-dimensional construct: The subscale *sharing* describes practitioners' attitudes towards sharing power, control, and information with their patients and the extent to which patients should be involved in decision-making, whereas the subscale *caring* considers practitioners engagement in psycho-social aspects of therapy and interest in their patients' expectations, wishes and life circumstances (*Krupat et al., 1999*). The response-format is a six-point approval scale, with higher values corresponding to stronger patient orientation (*Krupat et al., 2000*). More patient-centered practitioners are shown to engage with patients rather on lifestyle issues than on biomedical information; on the other hand, their patients are more willing to share information and to engage with the doctor (*Shaw, Woiszwillo & Krupat, 2012*). *Kiessling et al. (2014)* introduced a German translation of the PPOS as a shortened 12-item scale (PPOS-D12). In their validation study, the authors

evaluated the psychometric properties based on two surveys with students of dentistry and human medicine (total $N = 396$). They adopted the factor-structure of *Krupat et al. (2000)*, *i.e.,* both the number of latent constructs and corresponding attributions of the manifest indicators to the latent constructs were identical to the original PPOS model. As evident from the reported parameters, in their study on PPOS-D12, *Kiessling et al. (2014)* took an explorative rather than confirmative approach by performing principal component analyses, presumably using a fixed number of two factors to be extracted to reproduce a two-factor structure. Thus, an evaluation of the construct validity of a German translation of the PPOS regarding the goodness-of-fit of the theoretically assumed factor structure to the actual observed data is still pending. In addition, the practice of excluding items strictly based on formal-statistical criteria can be criticized, as it leaves the scale with a number of redundant items. Grammatical redundancy leads to inflated alpha reliability without adding to the conceptual representation of the latent construct (*Cortina et al., 2020*). In the present study, we therefore intend to develop an economical scale with a reduced number of items, that can be used time-efficient in everyday clinical practice, but nevertheless covers the dimensions *sharing* and *caring* as components of patient-centeredness.

The PPOS was developed to provide a measure to compare rating agreements between patients and practitioners (*Krupat et al., 2000*). Patient-practitioner interactions are inherently asymmetric due to the knowledge gap between practitioner (medical expert) and patient (medical layperson); as a result, practitioners and patients arrive at divergent judgments regarding patient-practitioner interactions, for example regarding discussions on (dis-)advantages of treatment options or the inclusion of the patient's perspective (*Floer et al., 2004*). The guidelines on patients' rights in Germany, for example, explicitly address this discrepancy in medical expertise by stating that patient-practitioner interactions must enable patients to assess the personal consequences of an envisaged treatment (*Federal Ministry of Health, 2007*). In this study, we use the knowledge gap between participants with and without previous medical experience as a proxy for the asymmetry between experts and laypersons in patient-practitioner interactions. Subsequent research compared PPOS-measures from male and female survey participants, medical students, doctors and allied health staff as well as corresponding to age and education (*Liu et al., 2019*; *Zhumadilova, Craig & Bobak, 2018*; *Wang et al., 2017*; *Kiessling et al., 2014*). All of these studies rely on the implicit assumption that levels of patient-centeredness measured are comparable across different groups—however, this assumption has never been tested. In order to close this gap, we developed a short version of the German translation of the PPOS. In this study we examine its psychometric properties and measurement invariance across participants with varying levels of medical experience and across gender.

## Measurement invariance as prerequisite for group comparisons

Measurement invariance is based on the assumption that distribution characteristics (*e.g.*, means or variances) from the operationalization of a construct have the same meaning across different groups of survey participants (*e.g.*, men and women), measurements over time (*e.g.*, in longitudinal studies) or different survey methods (*e.g.*, online and telephone surveys) (*Kline, 2016*). Measurement invariance is prerequisite in order to

attribute different measurement outcomes to actual differences between groups instead of differences in the measurement attributes (*Steinmetz, 2013*). In other words, measurement invariance assumes that a questionnaire or scale will function similarly across participants, irrespective of differences in their characteristics. With regard to the comparison of measurements in patients and practitioners, it must therefore be ensured that differences in the scores measured in both groups are due to actual differences in attitudes between patients and practitioners and not, for example, due to the fact that medical laypersons (patients) interpret items differently than medical experts (practitioners). Multigroup confirmatory factor analysis (CFA) is a common method to test for measurement invariance across groups (*Greiff & Scherer, 2018*) by comparing model fit-indices of factor models with increasing equality restrictions on parameters in order to achieve different levels of invariance like configural, metric, scalar and strict invariance (*Van De Schoot, Lugtig & Hox, 2012*; *Steinmetz et al., 2009*; *Meredith, 1993*): Configural invariance is given with identical numbers of factors and loading patterns across groups, assuming the corresponding manifest items load on the respective latent factors in all groups. Metric invariance is a supplement to configural invariance, if in addition, factor loadings are identical across groups. Invariant factor loadings are a prerequisite for comparing the relationships of factors between the two groups. Scalar invariance subsequently constraints item intercepts to be comparable across groups. Scalar invariance is a prerequisite for the comparison of latent factor means. Strict invariance assumes residual variances not explained by the factors are equal across groups. Strict invariance is a precondition for comparing indices of observed item scores. As strict invariance was described as an assumption that is rarely accomplished in applied research (*Van De Schoot et al., 2015*), partial invariance was introduced (*Van De Schoot, Lugtig & Hox, 2012*): Partial invariance is assumed if equality constraints are removed at a given invariance level in order to improve model fit, while at the same time the violation of the theoretically assumed structure is considered to be acceptable.

## MATERIALS & METHODS

### Data collection and participants

The data for this study were obtained in a project on medical students' attitudes toward the use of medical coercion. For this purpose, a cross-sectional survey of all first semester students of a medical faculty at a German university was conducted. In winter semester 2018/2019, a total of 369 students participated in the compulsory course "Medical Terminology". All students were invited to participate in the survey at the end of the course. They were informed that participation was voluntary and anonymous. No written consent was obtained. According to the medical faculty's Ethics Committee (EK 117/21), there were no ethical or professional objections to the study. A total of 332 students (human medicine, $n = 269$; dentistry, $n = 35$; logopedics, $n = 20$; doctoral students, $n = 7$; one student declined to indicate the program of study), completed the survey, which resulted in a 90% participation rate. In order to compare our results with previous research, we followed the approach of *Kiessling et al. (2014)* and included only students of dentistry and

human medicine. The following analyses are therefore based on a sample of 290 students (71.4% female, $n = 207$; deletion of 14 students (4.6%) with missing values). According to *Rubin (1976)*, missing not at random (MCAR) is a prerequisite for list wise deletion (only used for exclusion rates <5%). For each of the six items used in the subsequent analyses, the proportion of missing values is $\leq 3\%$. According to Little's MCAR test ($\chi^2 = 27.572$, $df = 27$, p $= .433$), we retain the null hypothesis of the data being missing completely random. The age of the respondents ranged from 18 to 36 years ($M = 21.7$, $SD = 3.7$). Almost half of the participants (47.6%, $n = 138$) had previous medical experience, *e.g.*, through paramedical trainings, internships or voluntary services.

## Measures

### German Patient-Practitioner-Orientation Short Scale (PPOS-D6)

The newly developedPPOS-D6 contains six items to be answered on a six-point approval scale ranging from 1 (= *I fully agree*) to 6 (= *I don't agree at all*). Of these, two sets of three items each are considered to represent *sharing* and *caring*, whereas the mean across all items is considered to represent *patient centeredness*, with higher scores reflecting more patient-centeredness. Starting from the German translation of the original scale (PPOS-D12, see table 2 in *Kiessling et al., 2014*), we eliminated semantically redundant items or items less related to the underlying concept in order to develop a short version of the scale (see Appendix 1): Among the items related to caring in PPOS-D12, the operationalization of personal relation in item 1 was conceptually represented with shorter wording in item 4, item 8 loaded on different subscales in two different cohorts and item 11 measured multiple dimensions, *i.e.*, cultural background and life situation. Among the items related to sharing, items 6, 7 and 12 seemed to represent obtaining information rather than participation in decision making. PPOS-D6 therefore included items 2, 3, 4, 5, 9 and 10 from PPOS-D12.

### Medical Experience

Respondents were asked whether they had already gained experience in the medical field before starting their studies (*e.g.*, through medical trainings, internships, voluntary services). Respondents were then divided into dichotomous groups with (= *1*) or without (= *0*) previous experience. From the open-ended responses to our question of what type of medical experience the respondents had previously gained, we know participants with medical experience had completed a voluntary social year related to clinical medicine, training as a nurse or training as a paramedic. As these trainings last one or more years, we assume that the knowledge gap between participants with and without previous medical experience is sufficient to serve as a proxy for the gradient in competences of medical experts *versus* layman.

### Staff Attitudes to Coercion Scale (SACS)

This 15-item questionnaire measures the extent to which medical staff consider the use of medical coercion as offending, as care and security or as treatment (*Husum, Finset & Ruud, 2008*). It comprises a six-point approval scale ranging from 1 (= *I fully agree*) to 6 (= *I don't agree at all*). We have recoded those items that were reversed in the original scale in order to calculate a mean index across all 15 items, with higher values indicating more

**Table 1** Mean values, standard deviations, skewness, kurtosis and zero-order correlations of study variables ($N = 290$).

|    | Variable | 1 | 2 | 3 | 4 | 5 | 6 | 7 |
|----|----------|---|---|---|---|---|---|---|
| 1  | Patient-centeredness | (.51) | | | | | | |
| 2  | Sharing | .87** | (.50) | | | | | |
| 3  | Caring | .69** | .23** | (.31) | | | | |
| 4  | Attitudes to medical coercion | .28** | .25** | .19** | (.78) | | | |
| 5  | Age | .10 | .12* | .02 | .03 | | | |
| 6  | Medical experience (yes = 1) | .12* | .08 | .11 | .02 | .43** | | |
| 7  | Sex (male = 1) | −.07 | −.08 | −.02 | -.24** | .19** | .13* | |
| 8  | Course (human medicine = 1) | .11 | .07 | .11 | .11 | .09 | .11 | .04 |
|    | Mean | 4.02 | 3.72 | 4.33 | 3.50 | 21.7 | – | – |
|    | Standard deviation | 0.59 | 0.88 | 0.60 | 0.51 | 3.68 | – | – |
|    | Skewness | 0.27 | 0.29 | −0.01 | 0.05 | 1.65 | – | – |
|    | Kurtosis | 0.20 | 0.09 | 0.48 | −0.05 | 1.82 | – | – |

**Notes.**

Cronbach's $\alpha$ in parentheses.

** $p < .01$.

* $p < .05$.

critical attitudes towards medical coercion. Cronbach's $\alpha$ for the total scale was .778. We used an ad hoc translation of the original English scale into German.

## Statistical analyses

To test the assumptions from the proposed measurement model, we used CFA to determine the consistency of the given factor structure with the data of our sample. As we intended to estimate standardized parameters of factor loadings for each item, we fixed the variances of the latent constructs uniformly. Model fit was estimated using root mean squared error of approximation (RMSEA), standardized root mean square residual (SRMR), comparative fit index (CFI) and Tucker-Lewis Index (TLI) in comparison to established cut-off values according to *Hu & Bentler (1999)* with RMSEA $\leq .06$, SRMR $\leq .08$ and CFI and TLI $\geq .95$.

We performed multigroup CFA for invariance testing across groups with varying levels of medical experience. CFI differences $\geq -.01$ between increasingly restricted models are regarded as an indicator of measurement invariance (*Little, 2013*). We performed CFAs using the lavaan package in R version 3.5.2 (*Rosseel, 2012*).

## RESULTS

Distribution parameters and zero-order correlations of study variables are reported in Table 1.

PPOS-D6 means for the overall scale as well as for the subscales *sharing* and *caring* are comparable to those reported by *Kiessling et al. (2014)* when using the extended PPOS-D12 scale (total scale 4.27; sharing 3.98; caring 4.56 according to *Kiessling et al. (2014)*). Both subscales show high positive correlations with the total scale for patient centeredness, *i.e.,* higher levels of *sharing* as well as higher levels of *caring* are associated with higher patient orientation. PPOS-D6 manifest item means ranged from 2.89 to 5.38 (*SD* between 0.78

and 1.36), skewness between $-1.26$ and $0.55$, and kurtosis between $-0.73$ and $1.80$ (see Appendix 2). Cronbach's $\alpha$ indicated poor internal consistency for the PPOS-D6 total scale and for both subscales. Inter-item correlations are low to moderate (see Appendix 2). Patient-centeredness is positively associated with critical attitudes towards medical coercion and medical experience. The subscales *sharing* and *caring* are both positively associated with critical attitudes towards medical coercion. There is also a positive association between the subscale *sharing* and age.

Since CFA estimations are significantly influenced by the manifest indicators' distributions and items did not hold the assumption of multivariate normal distribution (Mardia's skewness $\chi^2 = 251,635$, $p < .001$; Mardia's kurtosis $\chi^2 = 3,648$, $p < .001$), we used maximum likelihood estimation with Satorra-Bentler scaled $\chi^2$-test statistic providing robust parameter estimations when distribution assumptions are violated (*Finney & Di Stefano, 2013*). Estimated factor loadings, standard errors and *p*-values for the two-factor solution are shown in Table 2.

With all factor loadings being significant and according to the fit statistics ($\chi^2 = 9.399$ (*n.s.*), $df = 8$, RMSEA $= .025$ [$CI = .000; .077$], SRMR $= .033$, CFI $= .982$, TLI $= .965$) the two-factor solution can be regarded as a good approximation to the empirical data. The standardized covariance of *sharing* and *caring*, i.e., their correlation, is $.60$ ($SE = .14$, $p < .001$). Standardized loadings range from $.27$ to $.57$, which indicates substantial correlations between items and factors. With the exception of item 3, all factor loadings are $\geq .40$, which indicates substantial correlations between manifest indicators and according latent constructs. Standardized loadings of items related to sharing range from $.44$ to $.57$, while standardized loadings of items related to caring range from $.27$ to $.43$. Despite this imbalance, CFA results show that the two subscales represent distinct constructs.

Table 3 shows the results of individual CFAs for students with and without medical experience and different levels of measurement invariance across these groups.

The results indicate good fit of the two-factor model for both groups. According to delta-CFI, configural and metric invariance can be confirmed, whereas scalar invariance was not established. To identify non-tenable constraints in the partial scalar invariance model, we checked for significant modification indices (MI) associated to each of the constrained intercepts. MI revealed a significant improvement in model fit by releasing the constraint intercepts on item 3 across groups. With $\Delta$CFI $= -0.012$ between the metric and the partial scalar models, we assume partial scalar invariance, when the intercepts for item 3 are freely estimated across groups. An additional $\chi^2$-test confirmed that the metric and partial scalar invariance models did not significantly differ in model fit ($\Delta\chi^2 = 3.070$, $df = 3$, p $= .381$). Starting from the adjusted scalar model, we estimated a strict invariance model with residuals constrained to be equal for participants with and without medical experience. This model led to a considerable deterioration in model fit and strict invariance was rejected. Again, MI revealed a significant improvement in model fit by releasing the constraint residual variances on item 3 across groups. Comparing the partial scalar and partial strict models resulted in $\Delta$CFI $= -0.016$. However, the $\chi^2$-test indicated that the difference in model fit is not significant ($\Delta\chi^2 = 6.175$, $df = 5$, p $= .290$). We thus assume partial strict invariance by freely estimating the intercepts and residual

**Table 2  CFA results for the two-factor solution of the PPOS-D6.**

| Factor | Item no. | Loading (SE) | p-value | Std. loading |
|---|---|---|---|---|
| Sharing | 2 | .78 (.11) | .000 | .57 |
| | 5 | .54 (.09) | .000 | .50 |
| | 6 | .55 (.11) | .000 | .44 |
| Caring | 1 | .44 (.12) | .000 | .43 |
| | 3 | .26 (.08) | .001 | .27 |
| | 4 | .31 (.08) | .000 | .40 |

Notes.
  SE, Standard Error; Std. loading, Standardized loadings.

**Table 3  Fit indices for single CFAs and measurement invariance models across medical experience.**

| Model | df | $\chi^{2a}$ | RMSEA | RMSEA 90% CI | SRMR | CFI | TLI |
|---|---|---|---|---|---|---|---|
| Medical experience ($n = 138$) | 8 | 9.108 | .033 | [.000; .112] | .048 | .974 | .952 |
| No medical experience ($n = 152$) | 8 | 9.641 | .036 | [.000; .105] | .048 | .950 | .906 |
| Configural Invariance | 16 | 18.721 | .035 | [.000; .089] | .042 | .964 | .933 |
| Metric Invariance | 20 | 22.176 | .028 | [.000; .079] | .047 | .972 | .958 |
| Scalar Invariance | 24 | 30.575 | .044 | [.000; .085] | .057 | .915 | .894 |
| Partial scalar invariance[*] | 23 | 24.252 | .020 | [.000; .073] | .049 | .984 | .979 |
| Strict invariance[*] | 29 | 36.236 | .043 | [.000; .082] | .074 | .902 | .899 |
| Partial strict invariance[**] | 28 | 30.468 | .025 | [.000; .071] | .057 | .968 | .965 |

Notes.
  None of the models is significant.
  [a] Satorra-Bentler corrected.
  RMSEA, root mean squared error of approximation; SRMR, standardized root mean square residual; CFI, comparative fit index; TLI, Tucker-Lewis index.
  [*] Intercept for item 3 freely estimated across groups.
  [**] Residual variance for item 3 freely estimated across groups.

variances for item 3 across groups. Comparing observed means in both groups showed that students with and without medical experience slightly but significantly differed in their levels of patient-centeredness (t = −2.057, $df = 283.166$, p = .041): students with medical experience had an observed mean score of 4.10 ($SD = 0.59$), students without medical experience had an observed mean score of 3.95 ($SD = 0.57$).

Fit indices of the two-factor model for male and female participants as well as fit indices of models with increasing equality constraints across gender are shown in Table 4. In the discussion section we outline why the following results should be interpreted with caution. After configural invariance was established, restriction of factor loadings across groups led to an improvement in model fit and metric invariance was confirmed. Introducing additional restrictions of item intercepts across groups led to considerable deterioration in CFI and scalar invariance was rejected. Again, we used MI to identify non-tenable constraints in the scalar model. As a result, we freed the intercept of item 4 across groups which led to an improvement in model fit and partial scalar invariance was confirmed. Additional restrictions on the residual variances to this model resulted in delta $\Delta$CFI =

**Table 4  Fit indices for single CFAs and measurement invariance models across gender.**

| Model | df | $\chi^{2a}$ | RMSEA | RMSEA 90% CI | SRMR | CFI | TLI |
|---|---|---|---|---|---|---|---|
| Male ($n=80$) | 8 | 5.898 | .000 | [.000; .104] | .048 | 1.00 | 1.12 |
| Female ($n=207$) | 8 | 15.345 | .067 | [.000; .118] | .050 | .836 | .692 |
| Configural Invariance | 16 | 21.482 | .049 | [.000; .097] | .044 | .929 | .866 |
| Metric Invariance | 20 | 24.020 | .037 | [.000; .084] | .049 | .947 | .921 |
| Scalar Invariance | 24 | 33.947 | .053 | [.000; .092] | .058 | .872 | .840 |
| Partial scalar invariance[*] | 23 | 25.287 | .026 | [.000; .075] | .051 | .970 | .961 |
| Strict invariance[*] | 29 | 32.057 | .028 | [.000; .074] | .061 | .957 | .956 |

**Notes.**
None of the models is significant.
[a] Satorra-Bentler corrected.
RMSEA, root mean squared error of approximation; SRMR, standardized root mean square residual; CFI, comparative fit index; TLI, Tucker-Lewis index.
*Intercept for item 4 freely estimated across groups.

$-.013$ between the partial scalar model and the strict invariance model. Assuming that this deterioration in fit is tolerable is supported by a $\chi^2$-test indicating that the difference in model fit is not significant ($\Delta\chi^2 = 6.732$, $df = 6$, $p = .346$). Thus, strict invariance was accepted. Female ($M = 4.05$, $SD = 0.54$) and male ($M = 3.96$, $SD = 0.69$) students did not significantly differ in their observed mean ratings of patient-centeredness ($t = 1.098$, $df = 117.99$, p $= .274$).

## DISCUSSION

In clinical practice, there is an increasing need for psychometric short scales that provide valid and efficient latent construct measurements in a short time. The present study contributes to this demand by developing a short scale and testing measurement invariance across participants with different levels of medical experience and across gender.

PPOS-D6 represents a good fit to the two-factor model with the dimensions *sharing* and *caring*. The results of our study show that the scale is a valid measure of attitudes towards the doctor-patient relationship, as it is associated with theoretically related constructs: as expected, patient-centeredness is positively associated with critical attitudes towards medical coercion, supporting the claim for acceptance of the patient's right to self-determination in shared decision making models (*Elwyn et al., 2012*). Future research might use additional concepts to support convergent and discriminant validity of PPOS-D6. In addition, PPOS-D6 produces partially strict invariant measurements for participants with and without medical experience. This is the first statistical evidence for PPOS measurements to represent the same latent construct across groups with different levels of medical experience. As (partial) strict invariance is a prerequisite for the comparison of observed means, this finding is particularly important for group comparisons in a clinical and therapeutic context, as these usually represent inherent competency gradients between doctors and patients and are the actual applied scenarios for which PPOS was originally developed.

Admittedly, the sum of many individual measurements (*i.e.,* more items) may lead to more precise representations of latent constructs (*Marsh et al., 1998*; *Emons, Sijtsma &*

*Meijer, 2007*). However, extensive scales and time-consuming surveys no longer fit the time restrictions of everyday clinical practice. Cronbach's $\alpha$ indicated poor internal consistency for the PPOS-D6 and for both subscales. Lower $\alpha$-levels are frequently reported for short-scales (*Schweizer, 2011*) and are a known shortcoming of other PPOS language versions (*Pereira et al., 2013*; *Mudiyanselage et al., 2015*; *Wang et al., 2017*; *Hurley et al., 2018*); as short-scales intend to reproduce the same factor structure as their long-scaled-equivalents, but at the same time measure latent constructs with less manifest indicators, the items of short-scales are more heterogeneous compared to their full-length equivalents and thus decrease inflation of internal consistency and inter-item correlations (*Cortina et al., 2020*). Regardless of internal consistency, short-scales can still provide equivalent measurements of the underlying latent constructs; emphasizing efficiency over consistency may therefore be acceptable for comparisons on group level rather than investigations of individual differences (*Ziegler, Kemper & Kruyen, 2014*; *Rammstedt & Beierlein, 2014*). Just as the original scale, the PPOS-D6 is intended for the former, *i.e.,* group comparisons between doctors and patients. Future studies should investigate test-retest-reliability in different samples in order to provide more appropriate reliability measures for short-scales.

With a number of $N = 290$ participants, our sample is quite small to achieve group level comparisons of equivalent group sizes beyond dichotomous categories. In addition, it is quite homogenous considering age, so we did not account for measurement invariance across age groups. With $n = 80$ male respondents in our sample, gender groups were quite small in order to provide an identifiable model that accounts for measurement invariance across gender. In order to find at least moderate non-invariant items, a rule of thumb for sample sizes is $N \geq 150$ for simple CFA models with normally distributed indicator variables and no missing data or 100 observations per group for multigroup modeling (*Wang & Wang, 2020*); with our test for measurement invariance across different levels of medical experience, we are just above these recommendations for minimal group size. However, the results on invariance across gender should be interpreted with caution. Although small gender groups did not lead to issues in model convergence, estimated standard errors may be biased. This could also be a reason why full strict invariance was rejected.

We show that PPOS-D6 measurements in participants with different levels of medical experience are comparable. These findings do not provide evidence to justify comparisons of doctor and patient PPOS ratings, but are rather an approximation of such conclusions. The data for this study were obtained from a cross-sectional survey of medical students. Thus, we cannot make any statements about the stability of the measurements over time. Finally, all reported results apply exclusively to the German version of the scale. We offer an ad hoc English translation for understanding purposes only (Appendix 1). Future research might therefore provide additional insights into PPOS-D6 psychometric properties across divergent samples or stability of measurements in longitudinal studies.

## CONCLUSIONS

We conclude that PPOS-D6 is a valid measure for patient-centeredness due to its psychometric properties and partial strict invariance across groups with different levels of

medical experience and gender. This short scale can be useful for different research contexts dealing with doctor-patient interactions and especially where time is a crucial constraint to research.

## ACKNOWLEDGEMENTS

The authors thank Edward Krupat for his helpful comments on a previous version of this manuscript and Claudia Kiessling and Wolf Langewitz for their permission to use PPOS-D12 during scale development. The authors also thank the reviewers for their valuable comments, which helped to improve this manuscript.

### Funding
The authors received no funding for this work.

### Competing Interests
The authors declare there are no competing interests.

### Author Contributions
- Roman Pauli conceived and designed the experiments, performed the experiments, analyzed the data, prepared figures and/or tables, authored or reviewed drafts of the paper, and approved the final draft.
- Saskia Wilhelmy conceived and designed the experiments, performed the experiments, authored or reviewed drafts of the paper, and approved the final draft.

### Human Ethics
The following information was supplied relating to ethical approvals (i.e., approving body and any reference numbers):

According to the Ethics Committee at the RWTH Aachen Faculty of Medicine, chair: Günther Schmalzing, there are no ethical or professional objections to the study (EK 117/21).

### Data Deposition
The raw questionnaire data, codebook, and translation of the PPOS-D6 items to English are available in the Supplementary Files.

### Supplemental Information
Supplemental information for this article can be found online at http://dx.doi.org/10.7717/peerj.12604#supplemental-information.

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
