# Peer review of "A short scale for measuring attitudes towards the doctor-patient relationship: psychometric properties and measurement invariance of the German Patient-Practitioner-Orientation Scale (PPOS-D6)"

_PeerJ, doi:10.7717/peerj.12604_

## Round 0.1 · original submission · Major Revisions

With the reviews from three independent reviewers in hand, I decided to ask for a major revision. Indeed, I reckon that the manuscript is clear, the study is well conducted, and overall, the statistical analyses are sound, even though the authors should make some amendments to make the study worth publishing.

The Introduction needs to discuss some more relevant literature further (See Reviewer #1), and more motivation for the study should be provided (see, for instance, reviewer #2). Also, the notion of measurement invariance should be better introduced to a broader audience (see Reviewer #1 and Reviewer #2).

The three reviewers concurred in pointing out that the choice of medical experience as a moderator was poorly motivated. Moreover, Reviewer #2 questions the appropriateness of this dichotomization.

Another major flaw that needs to be amended is the gender comparison without evidence of measurement invariance based on gender, as noted by reviewers #2 and #3.

Reviewer # 3 makes some relevant observations on the factor loadings on the sample size that seem to be relevant.

In general, I recommend the authors take a close look at the reviewers' comments and address each of them.

·

Basic reporting

Literature reference (line 100 ..): Please add reference to the seminal contributio of Meredith (e.g., 1993)
Meredith, W. (1993). Measurement invariance, factor analysis and factorial invariance. Psychometrika, 58, 525-543.

Literature reference (line 110): Partial scalar is not a level of invariance per se. You may have partial metric, partial scalar, or partial strict. Please add strict invariance. Please add a short paragraph explaining what is the meaning of invariance testing and what are the allowed across-groups comparisons according to the different invariance levels reached.

Experimental design

Research question: It is not clear to me why medical experience would moderate Patient-Practitioner orientation. Please elaborate more on this and give a better rationale for why medical experience is a moderator.

Validity of the findings

Testing for partial scalar invariance (lines 213-220): It is not clear to me what you did. Once you established that delta-chi or delta-cfi evidenced a worsening of fit, please check for significant Modification Indices (MI) associated to each of the constrained intercept. Signigicant MI give evidence of non-tenable constraint. Then you have to relax the non-tenable constraints at once, untile the remaining constraints are all tenable. Please better clarify the approach you followed

Comparisons of observed total scores across medical experience: Partial scalar invariance allows for comparison of latent means. In order to compare observed means through a statistical test like you did you have to provide evidence of at list partial strict invariance, that implies testing equality of residual variances (DeShon, 2004, Psychology Science, Volume 46, 2004 (1), p. 137-149). If residual variances are difference, you may different variability in scale variance across groups and this may influence the results of the t-test.

Gender comparisons (lines 223-224): Why doing gender comparison when you did not test for gender invariance ? Following your premises you have to test to gender scalar invariance before doing this comparison. Anyway, the same reasoning I did for comparison of observed total scores related to medical experience applies here: so, provide evidence of at least partial strict invariance across gender before comparing males and females. If strict invariance does not hold, then provide evidence of scalar invariance but compare latent means.

Additional comments

The paper is well written but presents relevant flaws in the data analytical approach. There are also imporant references to the literature that are lacking.
I suggest the authors to better clarify the rational of the study, and in particular why medical expertise is relevant as far as the difference in PPOS psychometric properties.

·

Basic reporting

GENERAL OVERALL COMMENTS

This paper presents data gathered from medical student in a medical faculty in German in the first semester during a compulsory course on "medical terminology" by administering a newly developed 6 itemed tool to assess patient centred attitudes; PPOS-D6 and another self-administered tool to assess attitudes towards medical coercion; Staff Attitudes to Coercion Scale (SACS) simultaneously. Validity and reliability of the PPOS D6 has been assessed by statistical analysis and correlating with medical coercion scale scores. However, results were not compared with validity and reliability of original PPOS or PPOS-12 that had been used in the same medical faculty earlier.

The second objective is to evaluate comparability or applicability of PPOS D6 score between two group of students "with previous clinical experience" and "without such experience" using measurement variance. This exploration may be useful specially because the PPOS is used to assess patient centred attitudes among doctors, nurses, medical students as well as patients. However simple categorising the group in to "with experience" and "no experience" may not be appropriate as experience may be just few days or several months. If the notion is that the total, sharing and caring values obtained using the same self-administered tool (PPOS) may not represent true patient centred attitudes in two different groups, the exploration should be designed with another robust and independent method assessment of patient centred characteristics simultaneously.

Experimental design

No comments

Validity of the findings

No comments

Additional comments

Back ground of the study will be helpful.

What was the reason for developing a new tool?
Was there any evidence or suggestions to indicate the need develop a shorter version of the PPOS?
What are the known PPOS score among study population?

The objective of the study is not clear. What was the objective?
Clearly defined objectives are always essential. Following objectives seems possible in this study
1. is it developing a shorter version of PPOS; PPOS-D6
2. Validating the PPOS -D6 already in use (still the question how did you developed PPOS - D6 remains)
3. Evaluate the comparability of PPOS among experienced and inexperienced students

Methodology

Methodology is not clear.

1. Stepwise process of developing PPOS - D6 should be explained.
2. How did you select those 6 items
3. You have selected PPOS item 5, 12 and 15 for sharing attitudes and 3,6, and 7 for caring attitudes. What is the basis?
4. Did you adopt a forward and backward translation method. (Email conversation with Edward Krupet has commented about it)
5. In the process of content validation comparison with other psychometric properties is a good idea. Researchers have used JSE, communication skills and FGD. You seem to use SACS. It would be better to introduce the validity of this tool briefly.

It is better to have clear and consistent understanding about the purpose of the PPOS; the key ward is patient centredness.

• The Patient-Practitioner Orientation scale (PPOS) is a measure often used in health-communication research to assess for patient and provider belief regarding patient centredness.
• Patient- centeredness is the practitioner’s treatment approach to the illness experience of the provider beliefs regarding patient-centeredness. patient, rather than simply treating the biomedically defined disease.
• The scale can also be broken into two distinct factors: sharing, and caring. The sharing factor indicates a respondent’s belief that the provider is oriented to share power in their medical-care relationship. The caring factor indicates a respondent’s belief that the provider is oriented to caring about the patient- provider relationship, the patient’s emotions, and has interest in the patient not simply the disease.
Reference
1. https://www.mededportal.org/doi/10.15766/mep_2374-8265.9501
1. Original Citation: Krupat E, Hiam CM, Fleming MZ, Freeman P. Patient-centeredness and its correlates among first year medical students. International journal of psychiatry in medicine. 1999;29(3):347-56.

Line 8 &9 - The objective of the original PPOS was NOT to " assess perception about the quality of consultation"
Line 35 & 36 - PPOS is NOT to assess congruence (coincide) in their interaction between doctors and patients
Line 55 & 56 - PPOS did NOT developed to model patient satisfaction as a consensus between doctor and patients.

Line 41, 42 & 43 - " PPOS values are comparable between different groups " NOT used as an assessment of individual. The purpose of the concept of "measurement invariance" is not clear. (I am not a statistician)
line 18 -19 - The statement presents as results (looks more like a conclusion) is not clear to me; results indicate that PPOS-D6 provides valid and reliable measurement of patient centredness that are invariant across participants with different medical experiences.

Line 21 - 22 - The conclusion is more like a recommendation. However, I feel whether a guarded statement " The tool should be further tested for its the suitability and efficiency to use for education research.

Line 26-27 - what are those four ideal models (Emanual)

Line 46 - 47 - Yes agree a shorter version can save time. Did you have the same concern in your unit?

Line 57 - 59 - concept of sharing and caring could be explained better.

" Caring refers to the extent of the respondent’s belief about the importance of emotions, good interpersonal relationships and treating the patient as a whole rather than as a medical condition during doctor patient encounters. Sharing reflects the willingness to share information and power with patients as well as willingness to share control in decision‐making."

Line 65 - Need correction (Typo)

line 65 - 69 - what is the objective of lengthy evaluation of the approach of Kiessling et al)

Line 82 - 88 - How did you select those 6 items, what was the basis?

Line 89 - 90 - The PPOS was developed to provide a measure to compare rating agreement between patients and practitioners.

line 93 -96 - It is not clear to me (probably to other prospective readers also) this notion about the issue of "comparison between groups". PPOS has developed to evaluate and compare caring and sharing attitudes among groups. NOT individuals. It is known that different groups of people have diverse attitudes. They are different and they change (evolve) their patient centred attitudes due to extrinsic or intrinsic factors.

line 95 - 96 - it is obvious PPOS D6 was NOT developed to resolve the issue with regards to comparability.

Line 138 - 139 - How did you developed PPOS-D6 ?

line 144 - 145 - How did you eliminate redundant items. Did you used data bank from Kiessling study

Line 147 - 148 - dividing in to two groups Yes and No with regards to experience may not be ideal as experience can be very short of very long

Line 151 - 157 - SACS needs a better introduction.

Line 155 - 157 - Not clear " we have recorded some items in order to calculate the
mean across 15 items ........................ Please rephrase. Is the SACS an English tool originally?

Line 160 - 170 - Statistical analysis should have focus on consistency of PPOS - D6 and would have been better to compare with PPOS (Krupat) and PPOS 12 that we understand. I am not a statistician to comment of detailed workout done here.

Line 180 - 182 - PPOS value are bound to differ as the sample would have been different. But more important comparison is alpha and factor loading.

Line 229 - 230 - How does the testing measurement invariance across participants helps in developing short and effective tool.

Line 235 - 237 - It is better to define what is "critical attitude towards medical coercion".

Line 271 - 172 - Do you believe that PPOS score obtained from patient does not reflect their patient centredness. To evaluate this, we have to design a study to assess patient centred attitudes using two or more method and compare between two groups.

Table 1 - Heading of table should be - Correlation between study variables; 1 - 7

According to my understanding, when you correlate same variable (sharing score with sharing score) correlation should 1. It is not so in this table. Why? Please clarify.

What you correlate is the total PPOS-D6 score, sharing score and caring score NOT sharing or caring as stated in the table.

Table 3 - Comparison of student "with medical experience" and "NO medical experience" has blurred due to additional statistical data. Difficult to comprehend for me. Do you see a difference between two groups.

·

Basic reporting

Clear and unambiguous, professional English was used throughout the article. Occasional typos are present, so I advise to take an action in this sense.
Examples:
Lines 213-214: The results indicate good fit of the two-factor model for both groups. According to delta-CFI, configrual

Line 41: Previous research explicitly assume

Literature references and sufficient field background/context are provided for the German translation of the scale. I would make more comparisons with published validations. For examples, low internal consistencies are reported also in other published versions (e.g.: Portuguese validation).

The structure of the article conforms to the format of ‘standard sections’ of the Journal.

Figures are relevant to the content of the article, of sufficient resolution, and appropriately described and labeled.

All appropriate raw data have been made available. However, I would add a sheet in the Excel file with labels to allow a faster replicability of analysis. The Authors have already shared the code-book so analysis could be replicated but inserting a sheet in the Excel file would make the replication more feasible and reduce subjective interpretation bias.

Experimental design

The study falls with the main aims and scope of the journal both from the point of view of the investigated area (health) and the type of article (scale validation). Tthe Authors presented a short version of the German Patient Practitioner Orientation Scale (Kiessling et al., 2014). The PPOS is a scale that was developed to provide a measure to compare rating agreements between patients and practitioners and so it is an important tool to use in health and clinical settings. The German validation and adaptation (Kiessling et al., 2014) of the original scale was composed of 12 items (PPOS-12). The Authors further developed the attempt to provide brief scales in clinical settings reaching a version composed of six items (PPOS-6). As the Authors already mentioned, in clinical practice, there is an increasing need for psychometric short scales that provide valid and efficient construct measurements in a short time. The Authors made proposed a scale in this direction and tested measurement invariance across participants with different levels of medical experience, as an approximation to group comparisons. Existing research compared male and female survey participants, medical students, doctors, and allied health staff as well as corresponding to age and education, assuming that levels of patient-centeredness are comparable across different groups. However, this assumption was not tested and so the Authors work is very important to contribute to reducing this gap. The Methods are described with sufficient details and information to replicate: raw data, ethical approval, permissions from Authors, translations, English code-books are provided.

Validity of the findings

All underlying data have been provided; they are robust, statistically sound, and controlled. In Table 1 I would add data about Skewness and Kurtosis (which you have discussed in the text). I have particularly appreciated the discussion of missing values imputations. I would discuss not only the significance of the factorial loadings on each sub-scale but also the values in absolute terms. There is an imbalance between sub-factors (items of the factor sharing are saturating on the factor from .54 to .57 in terms of standardized factor loadings, while items of the factor caring are saturating from .27 to .43). I would provide inter-item correlations to complement the low Cronbach Alpha values registered. I would provide information also about the total score and how the sub-factors are in relation to the total score.
Conclusions are well stated, linked to original research question and limited to supporting results. Limitations of the study are clearly reported. The Authors stated that "in order to find at least moderate non-invariant items, a rule of thumb for sample sizes is N ≥ 150 for simple CFA models with normally distributed indicator variables and no missing data or 100 observations per group for multigroup modeling (Wang & Wang, 2020); with our test for measurement invariance across different levels of medical experience, we are just above these recommendations for minimal group size." If the medical experience group is composed of 138 participants, how you have met these recommendations? Can you describe as a limit?

---

## Round 0.2 · Minor Revisions

Although the manuscript has substantially improved, Reviewer #1 has some remaining concerns that need to be addressed.

·

Basic reporting

This is the re-review of a previous paper that I reviewed. Please make reference to my previous review for details.
This new version impements almost all the suggestions I made.
Regarding this section I acknowledge that authors elaborated better the concept of invariance and included the reference to the seminal work of Meredith (1993)

Experimental design

This is unfortunately the section were my suggestions were not considered.
I report here what I wrote in the previous review:

Research question: It is not clear to me why medical experience would moderate Patient-Practitioner orientation. Please elaborate more on this and give a better rationale for why medical experience is a moderator.

Validity of the findings

Authors clarified better the process they followed in testing for partial invariance
Authors tested carfully for gender invariance before testing for differences in observed means

Additional comments

I acknowledge the efforts of the authors regarding the data analytical approach
that now is not more flowed.
Still there is this point that must be addressed:
I suggest the authors to better clarify the rational of the study, and in particular why medical expertise is relevant as far as the difference in PPOS psychometric properties

---

## Round 0.3 · Minor Revisions

I think that the authors now have addressed the main remaining points from the Reviewer.

I just ask the authors to check the accuracy in reporting your results. For instance, sometimes they report p < .000, which of course is incompatible with the axioms of probability. Other times, the authors report p >.000, which is a truism.

---

## Round 0.4 · accepted · Accept

I am pleased to inform the Authors that the manuscript is now deemed suitable for publication.